# Transient interactions drive the lateral clustering of cadherin-23 on membrane

Cheerneni S. Srinivas[1,4], Gayathri S. Singaraju[1,4], Veerpal Kaur[1,4], Sayan Das[1], Sanat K. Ghosh[2], Amin Sagar [1], Anuj Kumar[2,3], Tripta Bhatia[2] & Sabyasachi Rakshit [1,3 ✉]

Cis and trans-interactions among cadherins secure multicellularity. While the molecular structure of trans-interactions of cadherins is well understood, work to identify the molecular cues that spread the cis-interactions two-dimensionally is still ongoing. Here, we report that transient, weak, yet multivalent, and spatially distributed hydrophobic interactions that are involved in liquid-liquid phase separations of biomolecules in solution, alone can drive the lateral-clustering of cadherin-23 on a membrane. No specific cis-dimer interactions are required for the lateral clustering. In cells, the cis-clustering accelerates cell-cell adhesion and, thus, contributes to cell-adhesion kinetics along with strengthening the junction. Although the physiological connection of cis-clustering with rapid adhesion is yet to be explored, we speculate that the over-expression of cadherin-23 in M2-macrophages may facilitate faster attachments to circulatory tumor cells during metastasis.

[1] Department of Chemical Sciences, Indian Institute of Science Education and Research Mohali, Mohali, Punjab, India. [2] Department of Physical Sciences, Indian Institute of Science Education and Research Mohali, Mohali, Punjab, India. [3] Centre for Protein Science Design and Engineering, Indian Institute of Science Education and Research Mohali, Mohali, Punjab, India. [4] These authors contributed equally: Cheerneni S Srinivas, Gayathri S Singaraju, Veerpal Kaur. ✉email: srakshit@iisermohali.ac.in

Cadherins predominantly maneuver the active cell-adhesion processes for both vertebrates and invertebrates. Two modes of binding are known for cadherins, trans-binding, and cis-binding. While in trans-binding, the terminal extracellular (EC) domains of cadherins from opponent cells interact in a hand-shake conformation with terminal EC domains[1,2], cis-binding is the lateral interactions among cadherins of the same cell[3]. Trans-interactions are responsible for mediating cell-cell adhesion. The cis-interactions are speculated to increase the molecular ordering and the stability of cell-cell junctions through clustering of the proteins on the cell membrane[4]. The trans- and cis-interactions function cooperatively to form the intercellular junction[5]. The trans-binding mode is well established from structural and kinetic studies in vitro as well as in cellulo localizations. The existence of cis-binding was first inferred from the widespread localization of cadherins at cell-cell junction[6,7]. Unlike trans-dimers, the direct existence of cis-dimers are not yet reported. One of the well-accepted proposals is that the trans mediates direct contacts among opposing cells via a diffusion (trap) kinetic approach and secures a cell-cell junction[8,9]. Approximately a cluster of five independent classical E(pithelial)-cadherin proteins diffuse across the membrane and form the trans-mediated cell-cell adherin junctions with neighboring cells[10]. Lateral or cis-interactions, thereafter, influence the clustering of cadherins on the premature junctions and strengthen the association between cells. Two specific cis-interactions are proposed from the crystal lattice of N-, C-, or E-cadherins that extend the intercellular junction linearly[5,11,12]. What was lacking in the proposal is the molecular pathway that laid the transition from a linear array to the two-dimensional arrangement at the cadherin mediated cell-cell junctions. For type I cadherins, it is proposed that a combination of cis and trans-interactions via exclusive shape complementarity form the two-dimensional adherin junction[5,13]. Another observation from the diffusion-dynamics of lateral clusters is the presence of nonspecific yet attractive interactions in addition to specific cis-binding that break the directional boundaries of interactions[14]. We propose that the transient and nonspecific intermolecular interactions can independently drive the two-dimensional arrangement of cadherins on cell membranes without any specific cis- or trans-interactions. Using a combination of biophysical and biochemical methods, we confirmed no existence of cis-dimers or higher-order oligomers of cadherins via lateral interactions.

Clustering of solute in a solution is a typical example of phase separation. In cell biology, such a liquid-liquid phase separation (LLPS) is common in the cytoplasm. It is the developmental origin of the membrane-less liquid compartments like nucleoli[15], centrosomes[16], Cajal bodies[17], and stress granules[18]. Relatively uncommon, but the existence of LLPS is also reported with proteins like Zonula Occludens[19] and nephrin[20] that are anchored to cell-membrane and mediate multiprotein cell-adhesion, signal transduction. Transient, weak, and favorable nonspecific interactions among like-neighbors are thermodynamically responsible for phase separations of biomolecules. Our hypothesis is that such favorable interactions independently can drive the cis-clustering of cadherins on two-dimensional confinement; without any trans- or stable cis-dimeric interactions.

Cadherin-23 (Cdh23) is one of the long non-classical cadherins with 27 EC domains. The two outermost EC domains of Cdh23 (Cdh23 EC1-2) from opposing cells interact homophilically in a trans-conformation and mediate strong cell-cell adhesion among tissues like the heart, kidney, muscle, and testis[21–23]. Similar to classical cadherins, the cellular junction of Cdh23 is continuous and engages in lateral clustering[21]. Apart from cellular adhesions, Cdh23 also interacts heterophilically with protocadherin-15

(Pcdh15), another non-classical cadherin, in the neuroepithelial cells of the inner ear and serves as gating-springs for mechanotransduction in hearing[24]. Similar to homophilic trans-dimerization at the cell-cell junction, the heterophilic trans-interactions at the tip-links engage only the two outermost domains of proteins (Cdh23 EC1-2 and Pcdh15 EC1-2). Here we aimed to solve the molecular mechanism of cis-clustering of Cdh23 and its functional benefits. We performed a combination of photoinduced reactions and membrane biophysics to verify the existence of cis-dimers of Cdh23 in clusters. Modulating the clustering of Cdh23 using chemical spacers and blockers, we further deciphered the nature of interactions that drive the lateral clustering of cadherins. We confirmed that transient, weak, and nonspecific interactions independently drive the clustering. Hydrophobic association dominates such transient lateral interactions.

Free-standing cis-clustering of Cdh23 independent of trans-binding prompted us to decipher the functional role of cis-clustering on cell adhesion. We measured a significant acceleration in the rate of cell-cell adhesions driven by cis-clustering. Disruption of cis-clusters by hydrophobic blockers drastically dropped the rate. Notably, while the toxicity of the phase-separated states has already been proposed for intrinsically disordered proteins[25], the fast-aggregation of cells is a demonstration of the functional implication of LLPS in cell adhesion.

## Results

**Cdh23 preferably remains as a monomer in the solution**. Cdh23 is an extensively elongated protein with 27 EC domains (Cdh23 EC1-27) and a molecular weight of ~315 kD. Commonly, EC domains of cadherins engage in cis- and trans-interactions. Therefore, we are only interested in the EC part of the protein and refer to the entire EC region of Cdh23 as Cdh23 EC1-27 and all other smaller variants with respective domain numbers. For example, Cdh23 EC1-2 represents the first two domains of Cdh23 from the N-terminal. Traditional methods such as native gel electrophoresis and chromatography always failed to detect the higher-order cis-complexes for Cdh23 and many other classical cadherins (Fig. 1a). The general argument is that weak binding-affinity is responsible for a 'no-show' of complexes in harsh experimental approaches. Here, we performed the photoinduced cross-linking of unmodified proteins (PICUP) to capture any stable or metastable dimers or oligomers of Cdh23 in the solution. PICUP is preferred for such identification due to its short reaction time, the ability to arrest short-lived assemblies, and most importantly, no alteration in the native proteins[26]. PICUP captures short-lived oligomers by forming radical-induced covalent bonds between interactomes. Followed by PICUP, we performed one-dimensional gel electrophoresis to obtain the stoichiometric distribution of dimer and monomer of proteins in equilibrium. To increase the detection ability to ≥3 ng/protein, we used silver staining of PAGE. We noticed a dimer band for Cdh23 in PICUP (Fig. 1b, Supplementary Fig. 1). Reportedly, Cdh23 has a strong affinity to a trans-homodimer[21,22,27]. Upon blocking the trans-interactions using an excess of Cdh23 EC1-2 in solution, we detected no trace of Cdh23-dimers in the SDS-PAGE even for 5 μM of protein, indicating either no specific interactions for cis-dimerization or a significantly poor affinity of Cdh23 toward cis-dimerization (Fig. 1b).

**Cdh23 undergoes LLPS**. Though Cdh23 does not form cis-dimers in solution, we observed a spontaneous phase separation (LLPS) of Cdh23 in the liquid phase from the solution. The fundamental difference in the nature of interactions in higher-order complexation and phase separation is the interaction

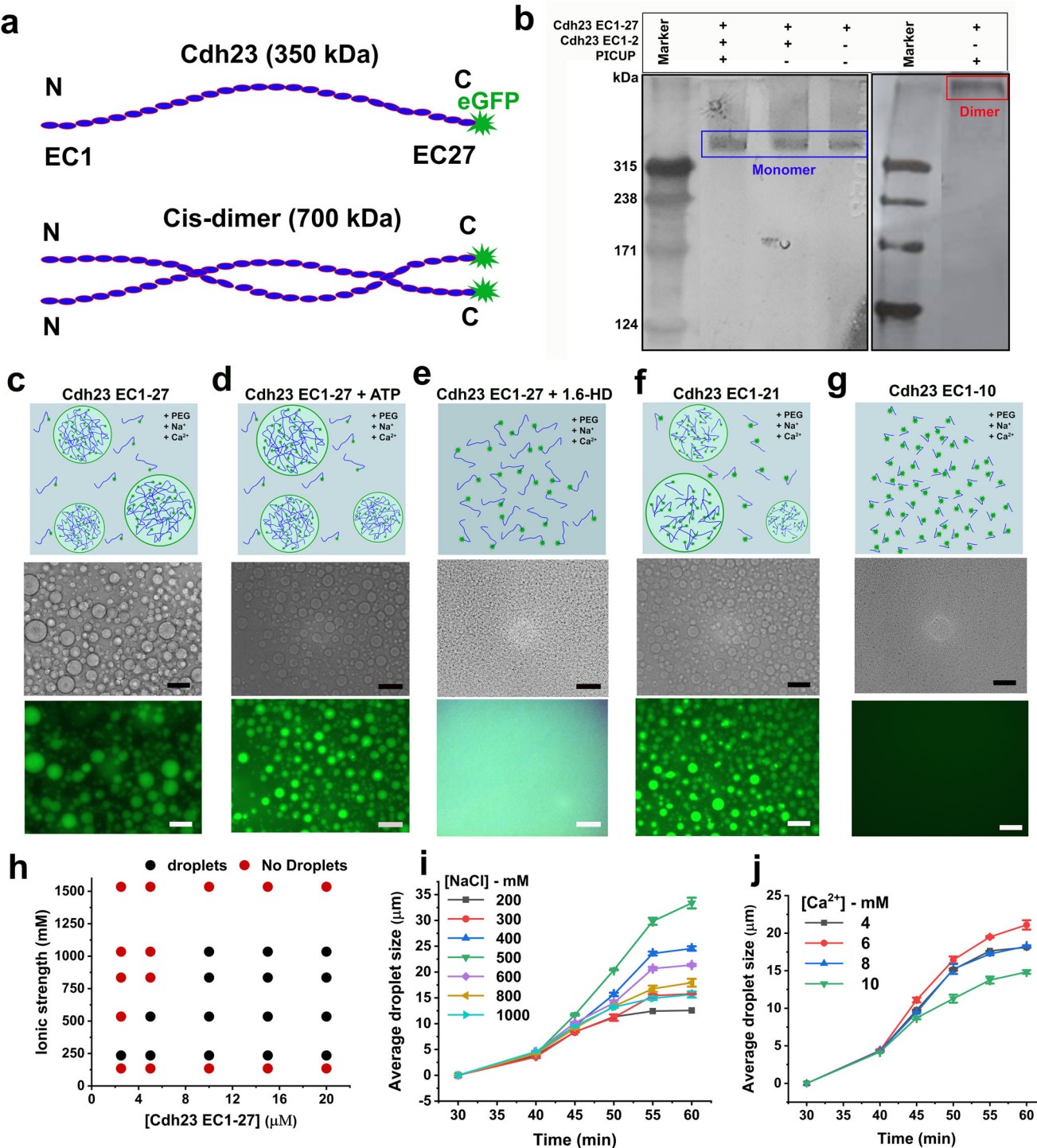

strength and specificity. While protein-protein complex formation relies on relatively stable and specific interactomes, phase separation of a protein in a solution is driven by weak, spatially distributed, transient intermolecular interactions. We observed a spontaneous formation of liquid droplets of Cdh23 EC1-27 above a critical protein concentration of 2.5 μM (Fig. 1c). The phase-separated state is stable over a range of ionic strengths (Fig. 1h). To visually track the phase separations in real-time under a fluorescence microscope, we recombinantly tagged eGFP (enhanced Green Fluorescent Protein) at the C-terminal of Cdh23 EC1-27 (Fig. 1a). Fusion among floating droplets, the gold standard for liquid condensates, is also noticed among Cdh23

condensates, indicating the fluid nature of the droplets (Supplementary Movie 1). Further from the time-trace analysis of the fusion events, we measured an average fusion-time of 5.0 ± 1.2 s, in range with other proteins that undergo LLPS[28–30] (Supplementary Fig. 2, Supplementary Movie 2).

Reportedly, Cdh23 mediates both homophilic and heterophilic trans-interactions using the outermost two EC domains (EC1-2). Two trans-interactions with two distinct binding affinities are reported between Cdh23 and Pcdh15[31]. The most robust trans-conformation with a dissociation constant of <1 μM was notified for the canonical variant. The second *trans*-conformation with a higher dissociation constant of 5 μM was observed for a truncated

**Fig. 1 Weak, nonspecific, and transient interactions among EC domains facilitate the clustering of Cdh23 in the solution. a** The schematic shows the monomer, and the hypothetical cis-dimer of Cdh23 EC1-27 recombinantly tagged with eGFP at C-terminus to track the association of cadherin-23 in solution and on membrane. **b** The silver staining of SDS PAGE gel shows the trans-dimer (~700 kDa) of Cdh23 EC1-27 when subjected to PICUP. However, only a monomer of Cdh23 is detected on blocking the trans interactions of Cdh23 EC1-27. The trans homophilic interactions of Cdh23 EC1-27 are blocked by incubating with Cdh23 EC1-2. Homophilic trans dimer of Cdh23 is the positive control. Blocking the trans-homophilic interactions using Cdh23 EC1-2 are used as 'No dimer' control. **c** The schematic shows the assembly of biomacromolecules in solution via liquid-liquid phase separation. Representative bright-field (upper panel), and fluorescence (lower panel) images of liquid droplet-like condensates of Cdh23 EC1-27 at 60 min. Scale bar: 50 μm. **d** The representative bright-field and fluorescence images of liquid droplets of Cdh23 EC1-27 treated with ATP. Scale bar: 50 μm. **e** The schematic shows the dissolution of liquid droplets with the solution. The representative bright-field and fluorescence images of liquid droplets of Cdh23 EC1-27 in the presence of 1,6-HD. Scale bar: 50 μm. **f** The representative bright-field (upper panel), and fluorescence (lower panel) images of liquid droplets of Cdh23 EC1-21 at 60 min. **g** Representative bright-field (upper panel), and fluorescence (lower panel) images indicate the inability of Cdh23 EC1-10 to undergo LLPS. **h** Phase diagram of liquid droplets of Cdh23 EC1-27 relating protein concentration and ionic strength of the buffer with droplet formation. **i** Growth kinetics of liquid droplets (μm) at a varying concentration of NaCl. Error bars represent the standard error of the mean (SEM) with $N = 20$ droplets. All the individual data points are shown in the scattered plots in the supplementary file. **j** The growth of liquid droplets (μm) of Cdh23 EC1-27 with time depends on $Ca^{2+}$ concentration. Error bars represent the SEM for $N = 20$ droplets. All the individual data points are shown in the scattered plots in the supplementary file.

**Table 1 Binding-affinities of the heterophilic and homophilic trans-complexes of Cdh23 as reported.**

| Homo/Heterophilic dimer | Interaction geometry | Method | T(°C) | $K_D$ (μM) | Reference |
|---|---|---|---|---|---|
| Cdh23 (EC1-2)-Pcdh15 (EC1-2) | Trans | Isothermal Titration Calorimetry (ITC) | 10 | 2.9 ± 0.4 | Nature, 2012 |
| Cdh23 (EC1-2)-Pcdh15 (EC1-2) | Trans | Surface Plasmon Resonance (SPR) | 25 | 0.84 ± 0.03 | Biochemistry,2018 |
| Cdh23 (EC1-2)-Cdh23 (EC1-2) | Trans | Analytical Ultra Centrifugation (AUC) | 20 | 18 ± 4 | FEBS, J.,2019 |

$K_D$ denotes the dissociation constant.

variant. It was, therefore, necessary to block the interference of the specific trans-interactions and measure the extent of droplet formation. Notably, the heterophilic trans-interaction with Pcdh15 has the highest affinity (Table 1)[22,23,32]. We accordingly blocked the trans-interacting sites of Cdh23 with ligand-protein Pcdh15 EC1-2. As a precaution, we first facilitated the heterophilic trans-interactions with an abundance of Pcdh15 EC1-2 (20 μM) in the experiment buffer and carefully altered the solution's ionic strength from an unfavorable phase separation condition to a favorable state via dialysis (Materials and Methods). We observed LLPS of Cdh23 EC1-27 and Pcdh15 EC1-2 complex, indicating that the droplets are predominantly due to interactions among EC domains other than EC1-2 of Cdh23 (Supplementary Fig. 3). Notably, the droplets of Cdh23 EC1-27 without Pcdh15 EC1-2 were more extensive than with Pcdh15 EC1-2, indicating that the additional trans-interactions contribute to the phase separation in vitro but are not essential for the LLPS. Overall, Cdh23 EC1-27 undergoes LLPS under physiological conditions, and the liquid droplets follow the characteristic feature of protein condensates.

**Nature of interactions driving LLPS.** What is the nature of interactions that facilitate the LLPS of Cdh23? Since the foundation of the dynamic condensed phase in a solution is transient and weak intermolecular interactions, the phase separated droplets rapidly and reversibly undergo mixing in the presence of interaction blockers. We used two different types of interaction blockers, Adenosine tri-phosphate (ATP) and 1,6-hexanediol (1,6-HD). ATP, being a condensed charge moiety, blocks the attractive electrostatic interactions for phase separation[33-35]. In contrary, 1,6-HD, an aliphatic alcohol, is known to weaken the hydrophobic interactions critical for LLPS and inhibit the condensation of solutes to liquid droplets[36,37]. When we treated the liquid droplets of Cdh23 with ATP, we observed no change in the propensity for phase separation (Fig. 1d). However, treatment of 1,6-HD disrupts liquid droplets of Cdh23 completely, indicating the predominant role of hydrophobic interactions in the LLPS (Fig. 1e).

Residue-wise analysis indicated a major population of non-polar residues, including Val (V), Isoleucine (I), Leucine (L), Glycine (G), Proline (P), and Alanine (A), spatially spread over several EC domains of Cdh23 including EC10, 11, 14, 16, and 25[38]. In contrary, Cdh23 EC1-27 also possesses 440 negatively charged and 222 positively charged amino acid residues distributed throughout its length. It is thus inferred that the favorable interactions that drive the phase separation of Cdh23 are delicately balanced between repulsive electrostatic and attractive hydrophobic interactions. To experimentally delineate, we systematically varied the ionic strength of the buffer and estimated the optimal phase separation conditions. Ions screen the electrostatic repulsion among opposite charges and thus, favor hydrophobic interactions. Accordingly, we used monovalent and divalent ions ($Na^+$ salts & $Ca^{2+}$ salts, respectively) and monitored the droplet growth rate for optimization. We first varied $Na^+$ ions keeping $Ca^{2+}$ ions fixed at 4 mM. We noticed a gradual increase in droplet growth with increasing $Na^+$ ions, reaching an optimum at 500 mM (Fig. 1i, Supplementary Fig. 4a). The overall phase separation of Cdh23 EC1-27 was noticed for 100 mM–1 M of NaCl. Next, we set the $Na^+$ ions to 500 mM, altered $Ca^{2+}$ ions, and obtained an optimal phase separation at 6 mM of $CaCl_2$ (Fig. 1j, Supplementary Fig. 4b). Overall, we observed that the screening of electrostatic charges did favor phase separation. Importantly, $Na^+$ and $Ca^{2+}$ ions beyond the salting-out range showed no LLPS for Cdh23 EC1-27.

To further substantiate the role of hydrophobic interactions in phase separation, we used two truncated variants of Cdh23, (i) Cdh23 EC1-21 and (ii) Cdh23 EC1-10. We obtained spontaneous phase separation of Cdh23 EC1-21 while Cdh23 EC1-10 did not undergo phase separation for a wide range of buffer conditions, including the conditions maintained for Cdh23 EC1-27 (Fig. 1f, g). No droplets for Cdh23 EC1-10 also indicate no active role of the crowding agents or the buffer conditions for the phase separations of the other two Cdh23 variants.

**No cis-dimers of Cdh23 in the lateral clusters on lipid membrane.** LLPS of biomacromolecules in a solution is the manifestation of

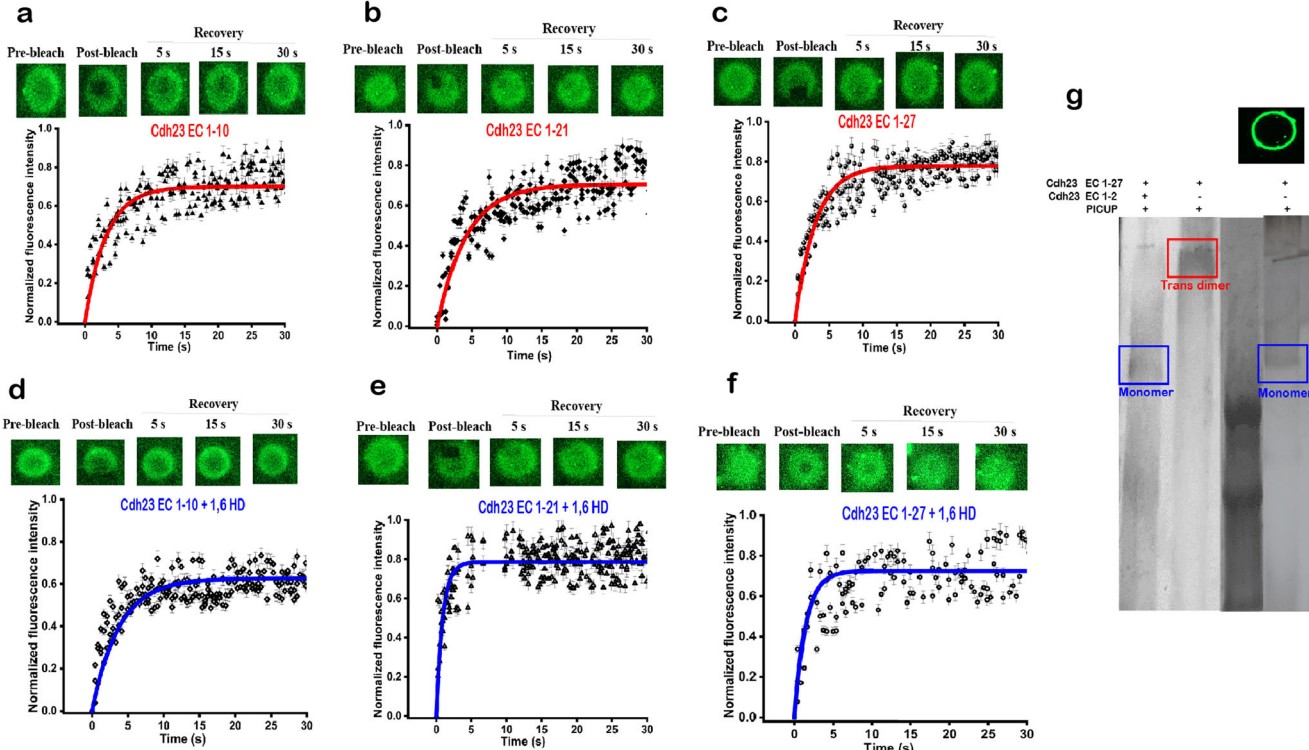

**Fig. 2 Cdh23 forms lateral clusters on GUV membranes. a–f** The fluorescence images of GUVs marking the pre-bleach, bleach, and post-bleach recovery regions at selected time points (upper panel), and the corresponding fluorescence recovery profile (normalized) (lower panel). **a–c** Are for GUVs anchored with Cdh23 EC1-10, Cdh23 EC 1-21, and Cdh23 EC1-27, respectively. **d–f** Are for the same GUVs anchored with Cdh23 EC1-10, Cdh23 EC 1-21, and Cdh23 EC1-27, respectively but after 1,6-HD treatment. The solid lines represent the fitting of the fluorescence recovery over time to the model (see methods). (**g**) PICUP on the clusters of Cdh23 EC1-27 on GUV membranes identifies only the monomer band in the silver stained SDS-PAGE. Middle panel is the repeat of Fig. 1b, indicating the relative position of the dimer band in red box. The upper inset shows the eGFP-fluorescent image of the representative GUVs used for PICUP.

weak and nonspecific transient intermolecular interactions. How effective is that transient interaction for lateral clustering on a membrane? Cadherins are trans-membrane proteins, and thus, the diffusion is restricted on two dimensions. Two-dimensional binding affinity is, therefore, significantly higher than the solution-based measurements. A dimensional restriction in diffusion and ordering in orientation enhances the binding affinity. It was thus speculative that the transient interactions could facilitate lateral clusters on a lipid membrane. To experimentally verify, we immobilized the eGFP modified Cdh23 on giant unilamellar vesicles (GUVs) and allowed free diffusion (See Methods). We noticed complete surface coverage of fluorescent proteins on GUVs. To estimate nonspecific attachment of proteins onto GUV membranes, we performed a control experiment with GUVs composed of only DOPC lipids. We incubated GUVs with proteins and monitored the extent of attachment from eGFP imaging (Supplementary Fig. 5). Absence of any eGFP signal on GUVs indicated negligible non-specific attachment of proteins.

Notably, Cdh23 mediates trans-homophilic interactions among neighboring cells at high-$Ca^{2+}$ [22]. Two N-terminal EC domains engage in the trans-interactions. $Ca^{2+}$ ions maintain a relatively stiff protein-conformations and make the terminal domains accessible for interactions. Accordingly, we noticed aggregations of Cdh23-attached GUVs at higher calcium contents (3 mM–6 mM) (Supplementary Fig. 6a), indicating the orientation of proteins with N-termini freely accessible for interactions. Chelating the calcium ions out from proteins using 1,2-bis(2-Aminophenoxy)ethane-N,N,N′,N′-tetra acetate (BAPTA) disrupted the GUV aggregates, re-instating that specific protein-protein interactions responsible for GUV aggregation (Supplementary Fig. 6b).

Fluorescence recovery after photobleaching (FRAP) has been a valuable tool for quantifying the clustering of molecules from the diffusion rates [39,40]. Accordingly, we performed FRAP experiments on GUVs where the C-terminal of Cdh23-GFP proteins was anchored via affinity (Fig. 2a–c) (See Methods). We performed the FRAP experiments at the polar region of GUVs for all variants of Cdh23 and recorded the slowest diffusion-rate of $0.124 \pm 0.005 \ \mu m^2/s$ for Cdh23 EC1-21 and the fastest for Cdh23 EC1-10 is $0.60 \pm 0.025 \ \mu m^2/s$ (Table 2). Also, we performed FRAP experiments at the periphery region of GUVs and noticed no significant differences in diffusion rates between polar and periphery regions (Supplementary Table 1). The diffusion-rate of Cdh23 EC1-27 is $0.232 \pm 0.007 \ \mu m^2/s$. The rate of diffusion is directly related to the cluster size. The slower the diffusion, the larger is the cluster size. Accordingly, Cdh23 EC1-21 forms a larger lateral cluster on the lipid membrane than the full-length Cdh23. This may not be surprising as CdhEC1-21 possesses a significant fraction of the long hydrophobic sequence. The smallest cluster size is formed with Cdh23 EC1-10.

Treatment of 1,6-HD that blocks the transient interactions enhances the rate of diffusion of proteins significantly to $0.97 \pm 0.049 \ \mu m^2/s$ and $0.94 \pm 0.054 \ \mu m^2/s$ for Cdh23 EC1-21 and Cdh23 EC1-27, respectively (Fig. 2d–f). Noticeably, both variants diffuse at comparable rates in 1,6-HD. We measured, however, no change in the diffusion-rate for Cdh23 EC1-10 w/o 1,6-HD (Table 2). Importantly, 1,6 HD did not affect the membrane integrity as the diffusion dynamics of adsolubilized Nile Red dye remained unaltered w/o HD (Supplementary Table 2). The treatment of 1,6 HD confirms that the weak, transient intermolecular interactions are predominant for lateral

**Table 2 Diffusion coefficients measured for the clusters of Cdh23 variants (with/without 1,6-HD) anchored to GUV membranes.**

| Cdh23 variants (w/o 1,6-HD) | Diffusion coefficient (µm²/s)[a] |
|---|---|
| Cdh23 EC1-10 | 0.60 ± 0.02 |
| Cdh23 EC1-21 | 0.12 ± 0.05 |
| Cdh23 EC1-27 | 0.23 ± 0.01 |
| Cdh23 EC1-10 + 1,6-HD | 0.60 ± 0.02 |
| Cdh23 EC1-21 + 1,6-HD | 0.97 ± 0.05 |
| Cdh23 EC1-27 + 1,6-HD | 0.94 ± 0.05 |

[a]It is important to note that the diffusion-coefficient of Cdh23 EC 1-27 and other smaller variants is closely dependent on the [Ca²⁺] ions in buffer.

clusters of Cdh23 on a membrane. Moreover, we tracked the fluorescence recovery over a cross-section and noticed the recovery two-dimensionally, indicating the lateral clustering is distributed over the GUV surface without any influence of trans-interactions.

Does clustering of cadherins on lipid membrane facilitate specific interactions among neighboring cadherins and generate cis-dimers or higher-order oligomers? To verify, we performed PICUP on lateral protein clusters on GUVs and monitored the existence of microstates from lateral interactions using SDS-PAGE of the PICUP samples. To our surprise, no higher-order bands appear in the PAGE, indicating the presence of no stable cis-dimers in the lateral cluster of Cdh23 on the lipid membrane (Fig. 2g). Overall, our study with GUVs confirms that transient nonspecific interactions purely drive the lateral clustering of Cdh23, and no stable cis-dimers with specific interactions exist or produce in the lateral cluster. Arguably, a dimer of individual tip-link proteins is already reported using an electron micrograph[41]. We also captured the cis-dimer of Pcdh15 using PICUP on GUVs, however, no Cdh23 cis-dimer. Further, our observations on Cdh23 contradict the existence of specific cis-dimer interactions in the lateral cluster of E-cadherins on the lipid membrane.

**Lateral-clusters accelerate cell-cell aggregation.** How does the lateral clustering of Cdh23 contribute to cells? Cadherins generally form anchoring junctions with the neighboring cells. Cdh23 is no different from the other family members and mediates vital cell-cell adhesion junctions in several tissues like the kidney, muscle, testes, and heart[21–23,42]. We, therefore, verified the effect of the lateral condensations of Cdh23 in cell-cell adhesion, more importantly, where lateral-clustering precedes the trans-interactions. We hypothesized that lateral clustering of Cdh23 on a membrane will increase the effective intercellular interacting interface and accelerate cell-adhesion. HEK293 is one of the widely used cell lines to study the functions of membrane proteins in cell biology. Further, these cells are relatively easy to transfect over other cell lines, and high transfection efficiency can be achieved. We have used HEK293 cells to verify the effect of lateral clustering of Cdh23 on cell-cell adhesion because the endogenous expression of Cdh23 in these cells is very low (Supplementary Fig. 7a). We thus monitored the aggregation of HEK293 cells exogenously expressing Cdh23 variants. Further, we measured the size of aggregates of the same cells but in the presence of 1,6-HD, where clustering of Cdh23 is blocked. As expected, we observed significantly rapid aggregations of HEK293 cells exogenously expressing Cdh23 EC1-27 than the cells transfected with Cdh23 EC1-10 (Fig. 3a, b, Supplementary Fig. 4c). We observed negligible aggregation with the untransfected cells. Treatment of 1,6-HD adversely affected the aggregation of HEK293 cells expressing Cdh23 EC1-27 (Fig. 3a, b). Importantly, cells in all the experiments formed Cdh23 mediated matured cell-cell junction after incubation, warranting the unaltered functionality of Cdh23 in the presence of 1,6-HD. HEK293 cells treated with siRNA for endogenous Cdh23 showed no aggregation within the experiment time.

Does cis-clustering on cell membrane depend on the extent of surface coverage by Cdh23? To check, we monitored the cell-aggregation among cancer cell lines, HEK293, HeLa, HaCaT, and A549 that express endogenous Cdh23 differentially. Our results from qRT-PCR (quantitative Real Time Polymerase Chain Reaction) and western blot, in corroboration with TCGA, indicate higher endogenous expression of Cdh23 in A549 and HaCaT cells and comparatively lower expression in HEK293 and HeLa cell-lines (Supplementary Fig. 7a). Amongst all in the list, HeLa has the least expression. We performed the cell-aggregation assays in the previously optimized buffer condition and noticed significantly faster cell-aggregations for A549 and HaCaT, than the low-expressing cell lines (HEK293 and HeLa) (Fig. 3c, d, Supplementary Fig. 4d). HeLa cells did not aggregate within the experiment time (Fig. 3d). To conclude the differences in aggregation based on the extent of the lateral-clustering, we performed the aggregation experiments with A549 and HEK293 cells in the presence of 1,6-HD. The aggregation of both cells dropped significantly in the presence of 1,6-HD and became comparable to HEK293 and HeLa cells (Fig. 4a, b, Supplementary Fig. 4e). Further, knock-out of Cdh23 in HEK293 and A549 cells using Cdh23-specific siRNA hindered the cell-aggregation significantly (Fig. 4a, b). To note, treatment of Cdh23-siRNA does not alter the E-cadherin expression (Supplementary Fig. 8). Overall, the kinetic data w/o 1,6-HD indicate that the differences in cell-adhesion of A549 and HEK293 cells are due to differences in the extent of lateral-clustering of Cdh23 (Supplementary Table 2). In addition, we estimated the mRNA expression levels of classical cadherins, E-cadherin and N-cadherin, in the mentioned cell lines using qRT-PCR (Supplementary Fig. 9). Similar to the expression patterns of Cdh23, A549 and HaCaT have comparable expressions of E-cadherin. However, A549 expresses N-cadherin orders higher than HaCaT. Expression of N-cadherin in HEK293 and HaCaT are comparable. Cell-adhesion kinetics of A549 and HaCaT are comparable too, indicating that at least N-cadherin may not be regulating the kinetics. To verify the effect of Cdh23 in cell-adhesion kinetics, we silenced the expression of Cdh23 in A549 cells using Cdh23-siRNA. The silencing of Cdh23 does dampen the kinetics significantly, indicating its active participation (Fig. 4a, b). When chemical treatments abolish the lateral-clustering, the individual Cdh23 molecules on the cell membrane follow the trans-mediated diffusion-trap model for cell adhesion, similar to classical cadherins. While the lateral-clustering increases the effective binding interface on a cell membrane and kinetically facilitates the cell adhesion, the diffusion trap is instead driven by the binding affinities between partners.

We further mimicked the differential extent of lateral-clustering of Cdh23 on GUVs. We titrated the Cdh23 EC1-27 attached GUVs with imidazole and monitored the change in clustering from the diffusion-rate. To note, the proteins are attached to GUV surfaces via Ni-NTA based affinity and treatment of imidazole replaces proteins from the GUV surfaces, thus reducing the extent of clustering. As expected, we first noticed a sharp rise in the diffusion-rate as we increased the imidazole from 0 M to 0.5 M (Fig. 4c). However, upon further increasing imidazole in solution, the diffusion-rate dropped gradually. We noticed significant drop in green fluorescent signal from the GUVs due to the loss of proteins from the surfaces. While a rise in diffusion can be attributed by faster diffusion of

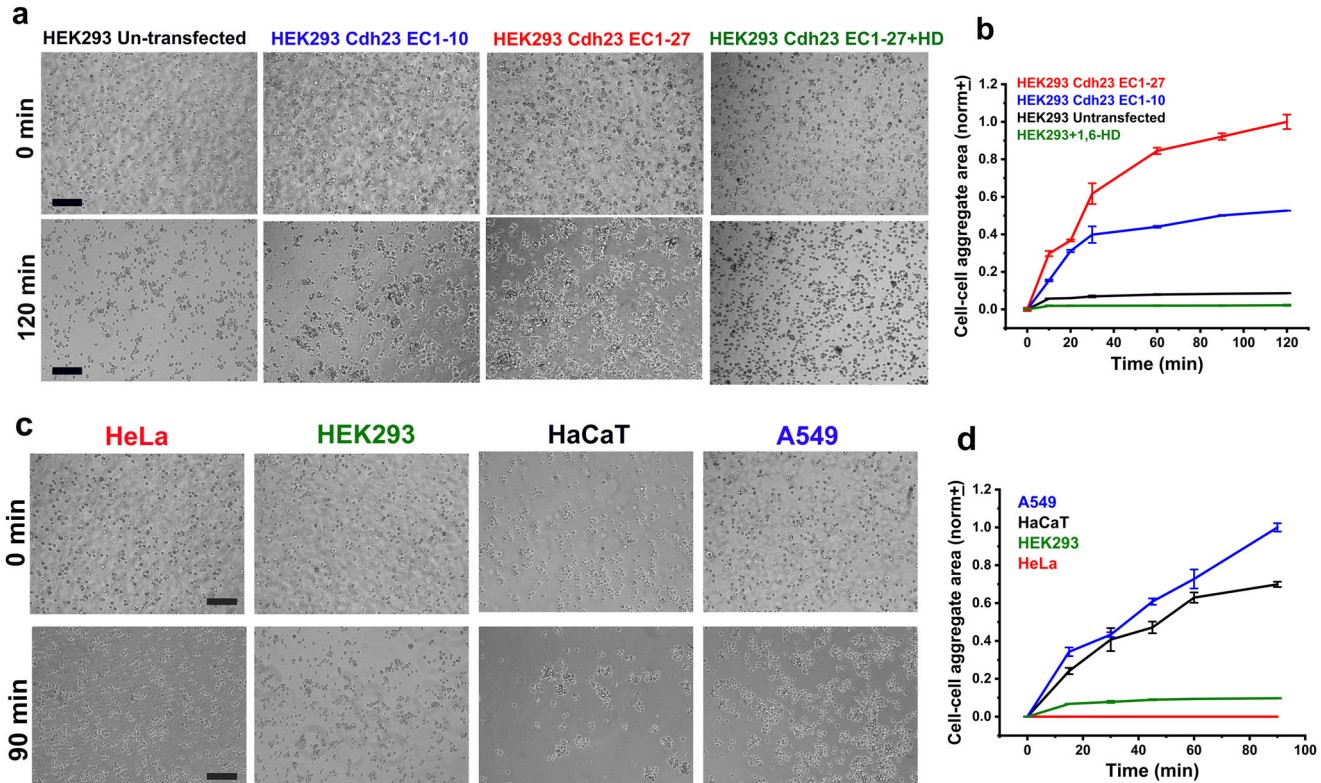

**Fig. 3 Lateral clustering of Cdh23 on the membrane enhances the kinetics of cell-cell adhesion. a** Time-stamp bright-field images of cell-cell aggregations of HEK293 cells untransfected (1st column), transfected with Cdh23 EC1-10 (2nd column), Cdh23 EC1-27 (3rd column), and Cdh23 EC1-27 along with the treatment with 1,6-HD (4th column). Scale bar: 50 μm. **b** The time-dependent growth of the cell-cell aggregation area of HEK293 cells exogenously expressing Cdh23 EC1-27 (red), Cdh23 EC1-27 and treated with 1,6-HD (olive), Cdh23 EC1-10 (blue), and untransfected cells (black). The error bars represent the standard error of the mean (SEM) with $N = 15$ aggregates. All the individual data points are shown in the scattered plots in the supplementary file. **c** Time-stamp bright-field images of cell-aggregates of HeLa, HEK293, HaCaT, and A549 cells differentially expressing endogenous Cdh23. Scale bar: 50 μm. **d** Growth of cell-cell aggregation area (in μm$^2$) with time for HeLa (red), HEK293 (Olive), HaCaT (black), and A549 (blue) cell lines. Error bars represent the standard error of the mean (SEM) for $N = 15$ aggregates. All the individual data points are shown in the scattered plots in the supplementary file.

relatively smaller clusters, a drop in diffusion-rate is a combining outcome of dilution and loss of clustering. A dilute surface inhibits lateral clustering and also allows proteins to travel long for fluorescence recovery. Importantly, treatment of imidazole did not alter the integrity of the GUVs as measured from the diffusion of adsolubilized Nile Red dye (Supplementary Table 2).

**Fluidic nature of Cdh23 clusters on the cell membrane.** To monitor the dynamics of Cdh23 at the cell-cell adhesion junction, we performed FRAP experiments on the intercellular junctions of HEK293 cells that are stably transfected with Cdh23. The protein was recombinantly tagged with eGFP at the C-terminal. We noticed localization of eGFP at the cell-cell junctions as expected for Cdh23 (Fig. 5a) and photobleached a confocal volume. Next, we monitored the fluorescence recovery along a line across the photobleaching spot (Fig. 5b, c). This is to identify if the recovery is from the new Cdh23 exported and recruited to the membrane by cells or diffusion of membrane-bound proteins. The fluorescence intensity profile across the line is expected to follow an inverted Gaussian profile with a deep at the center of the photobleaching spot (Fig. 5d). We noticed a widening in the Gaussian profiles with recovery and characteristics of diffusion of active proteins from the surrounding membrane. Recruitment of new proteins, in general, recovers the fluorescence intensity without diluting the surroundings, thus with a little widening of the Gaussian width[43]. Next, we plotted the width of the Gaussian ($\sigma^2$) with time-lapsed after photobleaching and fit to the linear equation and estimated the diffusion-coefficient of Cdh23

clusters at the cell-cell junctions (Fig. 5e) (Materials and Methods). The diffusion-coefficient of Cdh23 clusters at the cell-cell junctions is $0.6 \times 10^{-3} \pm 0.1 \times 10^{-3} \, \mu m^2 \, s^{-1}$, 8-fold slower than the reported diffusion-coefficient of classical E-cadherin ($D_{eff} = 4.8 \times 10^{-3} \pm 0.3 \times 10^{-3} \, \mu m^2 \, s^{-1}$) clusters of ~1000 molecules[43]. Overall, our FRAP data indicates that Cdh23 at the cell-cell junction is fluidic and diffuse in clusters.

## Discussion

At least two different molecular models are reported on the cis-clustering of classical cadherins. One of the models refers that the specific lateral intermolecular interactions among the EC domains drive the linear cis-clustering. The linear cis-interactions are, thereafter, expanded two-dimensionally at the junctions via intercellular trans-interactions[5]. The other model, however, involves isotropic nonspecific interactions in addition to specific cis-dimer interactions to explain the two-dimensional spread of cis-clustering at the cell-cell junction[14]. It is, therefore, a long standing challenge in cell-cell adhesion to decipher the nature of interactions that spread a two-dimensional cell-cell junctions. We showed that weak, transient, nonspecific, and spatially distributed multivalent interactions that drive LLPS of molecules in a solution, can be responsible for the two-dimensional spread of Cdh23 clusters on a membrane even prior to any trans-interactions. The interactions are predominantly hydrophobic in nature.

Further, the cis-clusters of cadherins are primarily known to strengthen the trans-interactions at the cell-cell junction[5]. Our

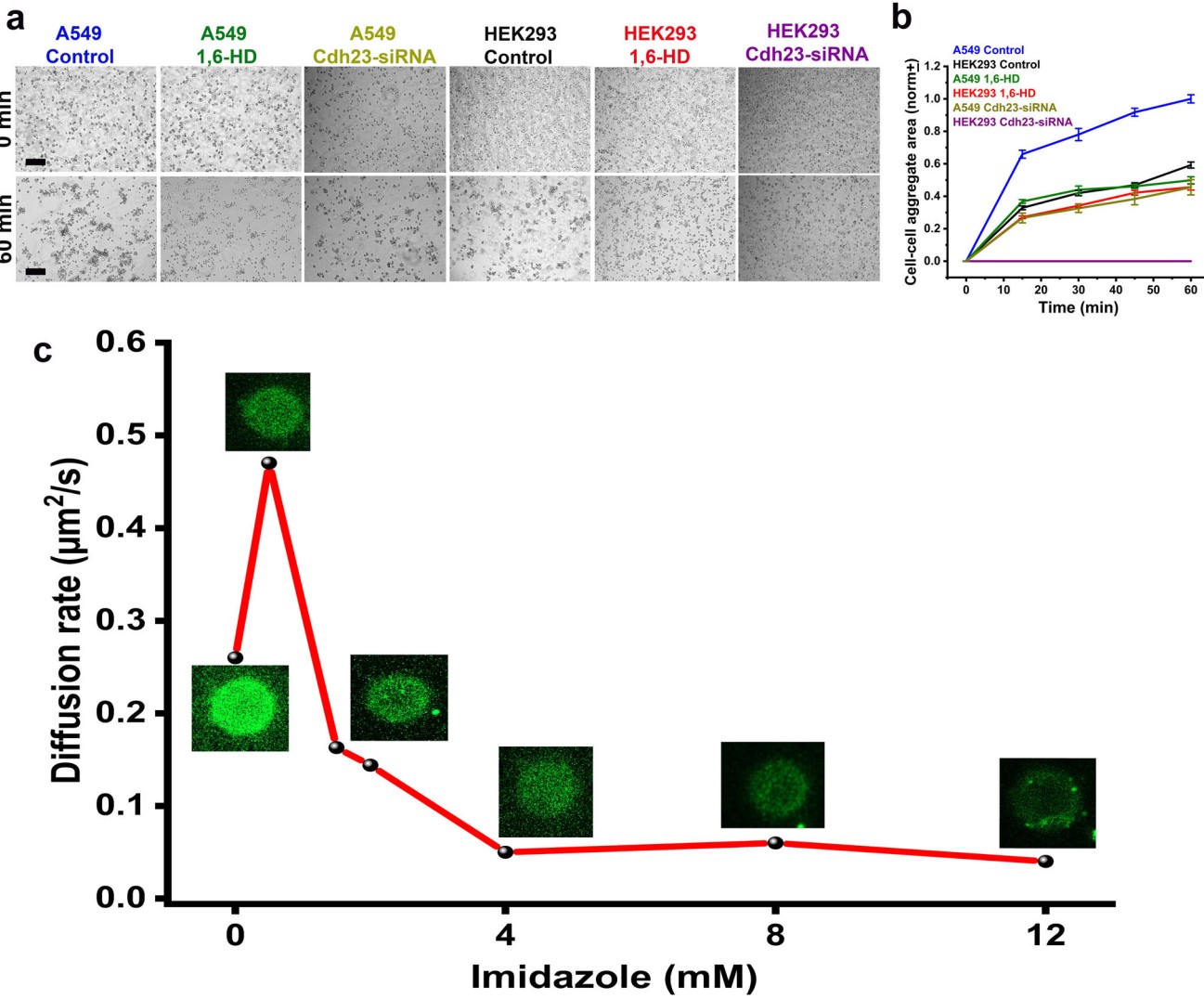

**Fig. 4 Effect of disrupted cis-clusters on cell-adhesion kinetics. a** Time-stamp bright-field images of cell aggregates of A549 and HEK293 cells in the absence (control) and presence of 1,6-HD, and treated with Cdh23-siRNA. Scale bar: 50 μm. **b** Growth of cell-cell aggregation area (in μm$^2$) with time for A549 cells in control (blue), treated with 1,6-HD (olive) and Cdh23-siRNA (dark yellow), and HEK2923 cells in control (black), treated with 1,6-HD (red) and Cdh23-siRNA (purple). Error bars represent the standard error of the mean (SEM) for $N = 15$ aggregates. All the individual data points are shown in the scattered plots in the supplementary file. **c** The gradual change in the diffusion-rates (μm$^2$/s) of Cdh23 EC1-27 anchored to GUV membranes by His-tag with increasing concentration of imidazole (mM).

experimental observations, however, identify the independent role of lateral clustering on cell-adhesion kinetics. We noticed that Cdh23 could condensate to clusters at the cell membrane independent of trans-mediated anchorage to neighboring cells and as super-adhesive, can rapidly seize the floating cells into aggregates. Though the implication of rapid cell-cell adhesion/communication in physiology or life sciences is not yet clear. Cdh23, among other cadherins, is significantly overexpressed in tumor-infiltrating M2-type macrophages[44] and microglia[45,46] (Supplementary Fig. 7b, Supplementary Fig. 10). M2-type macrophages associate with the circulatory tumor cells (CTCs) on the go and help in metastasis[47,48]. A quick cell-cell adhesion is thus essential in this process. Though speculative, the fast adhesion between M2 macrophages and CTCs is facilitated by the condensed Cdh23 droplets.

The distinctive features of nonspecific and transient lateral intermolecular interactions, stretchable and tunable to different sizes and shapes, may be helpful in cell-cell junction, which routinely experiences mechanical assault. Our results address the

molecular cues that spread the lateral clustering of cadherin-junction in two-dimensions and decipher the kinetic control of cadherin junctions. Identifying the physiological or pathological phenomena that strongly depend on the rapid cell-cell communication and adhesion may open up another exciting field in cell-adhesion.

## Methods

**Cloning of domain deletion mutants of Cdh23.** The full-length Cdh23 (NP_075859), consisting of 27 EC domains, a transmembrane domain (TM), and a cytoplasmic domain (CD), was a generous gift from Dr. Raj Ladher, NCBS, Bangalore. Using this construct, we recombinantly generated domain deletion mutants. We have subcloned the same construct in pcDNA3.1 (+) plasmid, which codes for Neomycin resistance. All the constructs were cloned between NheI and XhoI restriction sites with (S)-Sortase-tag (LPETGG)-(G)-eGFP-tag and (H)-His-tag; SGH-tag at downstream (C-terminus of the protein) in the same order. All the recombinant constructs were verified through double digestion, PCR amplification, and DNA sequencing.

**Protein expression and purification.** All recombinant Cdh23 variants for in vitro studies were expressed in the ExpiCHO suspension cell system (A29129

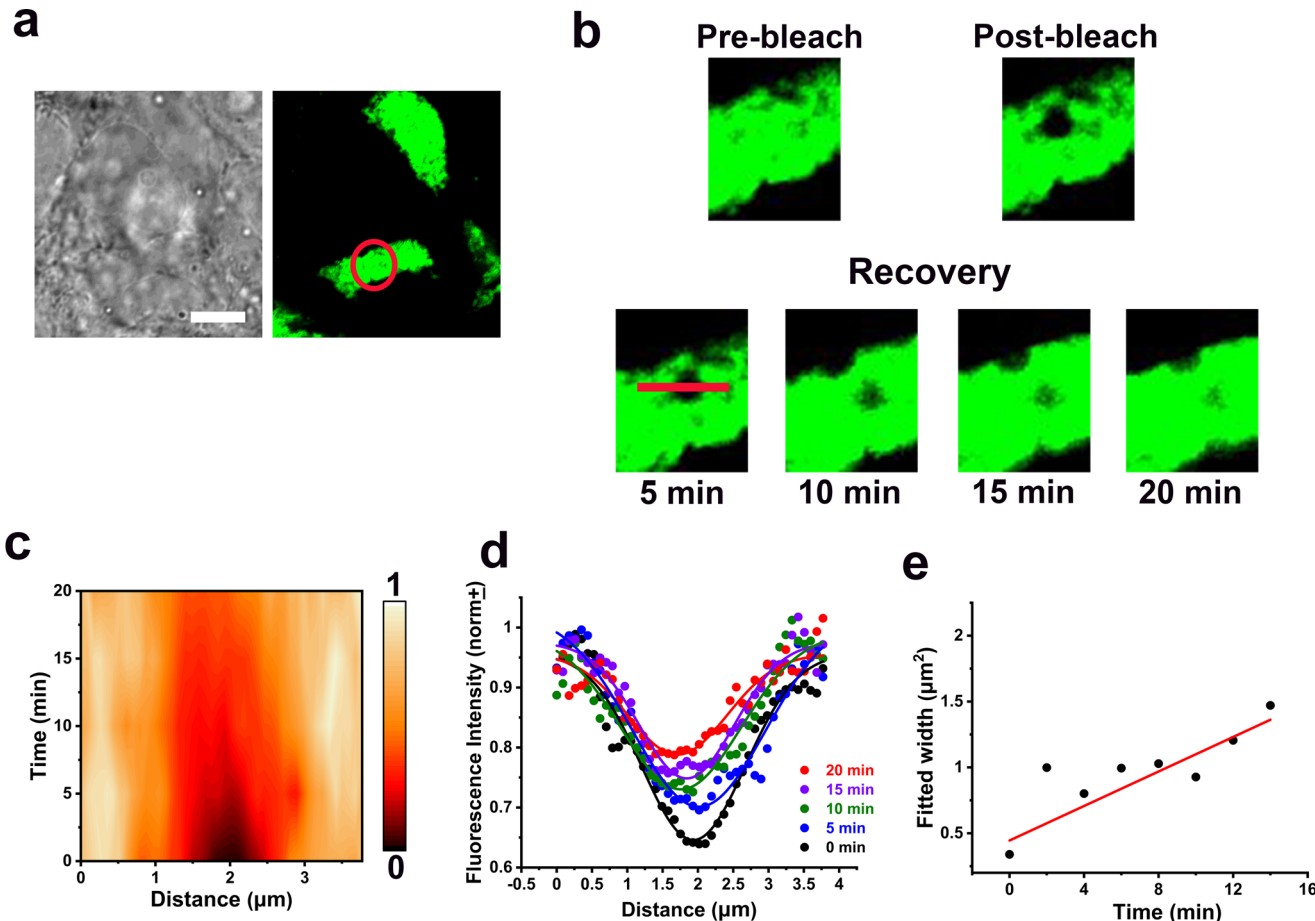

**Fig. 5 FRAP to probe the fluidic cis-clusters of Cdh23 at the cell-cell junctions. a** Representative bright-field and fluorescence images show the localization of Cdh23 at the cell-cell junction of HEK293 cells exogenously expressing Cdh23. Cdh23 is recombinantly tagged with eGFP at C-terminus. The circle (red) indicates the confocal volume for the FRAP experiment. Scale bar: 10 μm. **b** The cross-sections of fluorescence images indicate the regions of pre-bleach, bleach, and post-bleach recovery at the selected time points. The fluorescence recovery is monitored with time along the red line. The length of the line is 4 μm. **c** The contour plot represents the spatiotemporal distribution of Cdh23-eGFP along the red line. **d** The fluorescence intensity profile along the red line is plotted with the recovery time points. The solid lines are the Gaussian fits. **e** The widths ($\sigma^2$) from the Gaussian fits are plotted with recovery time. The solid red line is the linear fit to the data to estimate the diffusion-coefficient. Error bars indicate the standard deviation obtained from the Gaussian fit.

ThermoFisher Scientific), following the prescribed protocol for transfection in ExpiCHO cells. After 7 days, the culture media was collected by pelleting down the cells at 2000 rpm for 15 min at room temperature. The media was then extensively dialyzed against the dialysis buffer for 48 h and intermittently changed the buffer every 8 h. The dialyzed media with proteins were purified using affinity chromatography using Ni-NTA columns. The purity of the samples was checked using SDS-PAGE. Finally, the presence of protein was confirmed using western blotting with specific antibodies against GFP, Cdh23, and his-tag.

**In vitro droplet formation assay**. All purified proteins were prepared in a buffer containing 20 mM HEPES, pH 7.5, 100 mM NaCl, and 4 mM CaCl$_2$. Before each experiment, the proteins were centrifuged at 15000 *rpm* at 4 °C for 10 min to remove possible nonspecific aggregates. Then proteins were adjusted to reach designated concentrations. Each protein mixture (14 μM for each component) was injected into a homemade chamber and imaged using a Leica microscope (Leica DMi8) using a 40× objective lens. The time-lapse images were taken under brightfield and fluorescence filters. All the assayed droplets were thicker than 6 μm in height, so the central layers of optical sections were chosen for quantification. Over 10 or more droplets were measured for each protein to generate the phase diagram of the condensed phase. The images were analyzed by ImageJ, and the quantification was performed by Origin software.

**PICUP**. About 5 μM of purified Cdh23 was added with a metal complex containing 15 mM sodium phosphate, pH = 7.5, 150 mM NaCl, 0.125 mM [Ru(bpy)$_3$Cl$_2$], and an electron acceptor, 2.5 mM of Ammonium persulfate (APS), followed by the irradiation of visible light for 0.5 s. With no delay, the reaction mixture was quenched with 7 μl of 4× SDS-gel loading dye and heated at 95 °C for 5 min. Next,

SDS-PAGE and silver-staining were performed consecutively to visualize the cross-linked product. Cdh23 EC1-2 (5 μM) was added to block the trans-homodimerization of Cdh23 during PICUP experiments.

**GUVs preparation**. The mixture of DOPC (850375 P, Sigma-Aldrich), DGS NTA Ni (790404 C, Sigma-Aldrich), and DSPE-PEG Biotin2000 (Avanti) lipid solutions dissolved in chloroform was used to prepare GUVs. We followed the electro-formation method to prepare the GUVs[49]. Briefly, the calculated amounts of lipids mixture (1 mM DOPC: 1 mM DGS NTA Ni) were spread on the plates and allowed the solvent to evaporate under vacuum. GUVs were harvested in a 350 mM sucrose solution. The formation of GUVs was visualized using phase contrast and fluorescence microscopy (Leica, Dmi8), where the GUVs were labeled with Nile red (1 μM), a fluorescent dye (72485, Sigma-Aldrich).

**Tethering GUVs to the functionalized surface and the specific attachment of the protein to GUV membranes**. Glass base petri dishes were extensively cleaned with piranha and KOH etching before the protein incubations. The glass surfaces were coated with 1 mg/mL of BSA protein (Albumin, biotin-labeled bovine, A8549 Sigma-Aldrich) by incubating for 2 h, followed by a wash with MilliQ water. Next, the surfaces were incubated with 0.1 mg/ml of Streptavidin stock (S4762, Sigma-Aldrich) for 1 h and then washed with water. The GUVs were incubated on the modified surface for 4 h to anchor them to the surface through the non-covalent interactions between biotin-streptavidin. The surface adhered GUVs were incubated with ~6 μM of GFP-tagged proteins at 4 °C for overnight. The protein constructs possess 6xHis and eGFP tags at the C-terminus whereas GUVs are constituted by a mixture of DOPC and DGS Ni NTA lipids. 6xHis-tag of proteins at C-terminus thus attach to surface-bound Ni$^{2+}$-ions via coordination chemistry.

**FRAP and analysis**. FRAP experiments were performed at the polar and periphery regions of the tethered GUVs using a super-resolution microscope (ZEISS LSM 980 Airyscan 2) at 63X magnification. The experiments were performed in the presence and absence of 1,6 hexanediol in the buffer.

The fluorescence intensity (normalized) at the selected region was measured for three independent experiments and plotted against the time. Recovery half lifetime ($\tau_{1/2}$) was estimated by the global fit of fluorescence intensity profile with recovery time using the following equation,

$$f(t) = A(1-e^{-\tau t}) \tag{1}$$

where A indicates the mobile fraction of proteins and

$$\tau_{1/2} = \frac{\ln 0.5}{-\tau} \tag{2}$$

The diffusion coefficient for Cdh23 localized on vesicle membranes was calculated using the equation

$$D = \frac{0.88 * w^2}{4\tau_{1/2}} \tag{3}$$

where $w$ is the radius of the region selected to study FRAP.

**LLPS disruption studies**. To disrupt the liquid-like clusters, the cells were incubated with 2% (w/v) of 1,6-Hexanediol (H11807, Sigma-Aldrich) for 1 h. Similarly, 10% (w/v) of 1,6-Hexanediol was used to disrupt the liquid droplets of Cdh23 EC1-27 in vitro.

**Western blot and qRT-PCR**. The adherent cancer cell lines HeLa, HEK293, A549, and HaCaT were obtained from NCCS, Pune. All cells were cultured in high glucose DMEM media (D1152, Sigma-Aldrich) containing 10% FBS and 5% $CO_2$. We followed the standard protocol[50] for the western blotting of the lysates from the mentioned cell lines. Cadherin-23 (HPA017232, Sigma-Aldrich and PA5-43398, Invitrogen), eGFP (A11122, Invitrogen), and His-tag (11965085001, Roche) antibodies were used at the concentration of 1 µg/ml to detect the proteins.

RNA from different cancer cell lines was extracted using RNA isolation kit (Bio Rad) and treated with DNAse using DNAse 1 kit (AMPD1, Sigma-Aldrich). cDNA synthesis was done using a cDNA synthesis kit (Bio Rad). qRT-PCR was performed with the primers probing Cdh23 using the real-time PCR system (CFX96 Bio Rad).

**Cell-aggregation assay**. After 30 h of post-transfection, the cells were washed gently with PBS and then resuspended in Hank's buffer supplemented with 10 mM $Ca^{2+}$ ions to a final cell count of $10^5$ cells. Hank's buffer behaves like an incomplete media maintaining the osmolarity of the solution with the cells avoiding any bursting or shrinking of cells throughout the entire duration of the assay. After resuspending, the cells were imaged with a bright-field filter at ×10 magnification using a Leica Inverted Microscope (Leica DMi8) over a time trace for 2 h. The images were collected at 10 min, 15 min, 30 min, 45 min, 60 min, and 120 min when all the cells aggregated completely. The cell-aggregates at five different surfaces were considered at each time point, and the sizes were measured from the bright-field images using ImageJ software (NIH, Bethesda). The image analysis for measuring the area of each aggregate was done in ImageJ software. The aggregates with at least five cells are considered for the analysis. The mean area of aggregates over five different focal positions was measured and plotted against time. The aggregate size was compared over varying domain lengths for Cdh23. We performed all cell aggregation experiments at fixed cell numbers. 1% (w/v) of 1,6-Hexanediol was added to Hank's buffer to disrupt the LLPS during the cell-aggregation assays.

**Live-cell imaging and FRAP analysis**. Stably expressing Cdh23 HEK293 cells grown for confluency on a 35 mm glass-base petri dish were used for imaging. A super-resolution microscope (Zeiss LSM980 Airyscan 2) was used to image the cells maintained at 37 °C and 5% CO2. FRAP was performed on a confocal volume of 1 µm diameter at the cell-cell junctions where localization of eGFP was noticed. ImageJ software was used to measure the fluorescence intensity profiles of the line segment of 4 µm drawn across the photobleached region (line-scan analysis). The fluorescence intensity profiles (normalized) at different time points were fit to the Gaussian function in origin software. The fitted widths obtained at different time points were plotted against recovery time, and fit to linear regression to estimate the diffusion coefficient from the slope[43].

**Statistics and reproducibility**. All the experiments were performed in triplicates. The data are represented as mean ± standard Error of mean along with the sample sizes mentioned in the figure legends.

**Reporting summary**. Further information on research design is available in the Nature Portfolio Reporting Summary linked to this article.

## Data availability

The source data underlying the graphs are available as Supplementary Data 1. All other data available from the corresponding author on reasonable request.

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

## Acknowledgements

This work was supported by the Core Research Grant (CRG) by SERB, Govt. of India. SR acknowledges and thanks Dr. Raj K. Ladher, NCBS, for generously donating the Cadherin-23 plasmid. SR acknowledges the financial support by the DBT/Wellcome Trust India Alliance and Indian Institute of Science Education and Research Mohali (IISERM) for setting up the facilities. G.S.S., C.S.S., V.K., and A.S. sincerely thank IISERM for financial support. S.D. thanks DBT/Wellcome Trust India Alliance for the Project Assistant Fellowship awarded to S.R. A.K. is grateful to DST Inspire for funding. We acknowledge the Department of Biological Sciences, IISER Mohali, for giving access to the super-resolution fluorescence microscope facility granted by SERB for the Fund for Improvement of S&T Infrastructure in Universities and Higher Educational Institutes (FIST) Program. We thank Mr. Taseen Ahmad for assisting in acquiring super-resolution microscopy data and Mr. Hemraj for helping us with the preparation of GUVs.

## Author contributions

S.R. has supervised the project. G.S.S., S.D., A.S., and A.K. did the cloning, expression, and purification. G.S.S. and C.S.S. recorded and analyzed droplet condensation and cell-aggregation experiments. C.S.S. did the live-cell imaging and FRAP experiments. T.B., V.K., and S.R. has designed the GUV experiments. T.B. has supervised the GUV experiments and the corresponding data-analysis. V.K. performed PICUP experiments. V.K., S.K.G., and T.B. performed GUV experiments and analyzed FRAP on GUVs. G.S.S., V.K., and C.S.S. made the figures. S.R. and C.S.S. wrote the paper. G.S.S., C.S.S., V.K., and S.R. edited the paper. C.S.S., G.S.S., and V.K. have contributed equally in this work.

## Competing interests

The authors declare no competing interests.
