## [Peer review File · Communications Biology]

Reviewers' comments:

Reviewer #1 (Remarks to the Author):

The work by Srinivas et al. describes the cis interaction of the cellular adhesion molecules cadherin-23. Cis and trans interactions are hypothesised to work cooperatively in the formation of cellular adhesions. While the molecular origin of the trans interactions between cadherins on opposing cells are well described, much less is known regarding the nature of cis interactions of cadherins on the same cell. In this work, the authors use a range of methods to establish the nature of the cis interaction of cadherin 23. The authors main claim states that transient, weak, and nonspecific interactions hydrophobic interactions drive clustering.

The paper is well written, the data well presented and the majority of the conclusions drawn from the data are justified. Before publication the following comments should be addressed by the authors.

1. The authors state in the text that the "The phase-separated state is stable over a range of ionic strength", however no figure is referenced that shows this specific data. Please include the reference in the revised text.

2. In figure H,I,J, the dependence of droplet growth on NaCl, and Ca²⁺ concentration is shown. Given the previous statement about droplet stability and ionic strength, there actually seems to be a quite complex relationship between droplet size and ionic strength. What is the origin of this relationship, especially considering the conclusion that hydrophobic and not electrostatic interactions are driving the weak interactions between Cdh23?

3. The authors use a minimal membrane system to investigate cis interactions with the 2D membrane environment, coupling the Cdh23 to the GUV membrane using Ni-NTA lipids. Can the authors comment on the orientation of the Cdh23? Presumably, given the lack of transmembrane domains, the molecule would have more degrees of freedom, increasing the likelihood of interactions with the membrane that may lead to the molecule lying flat on the membrane. Can the authors rule this out and how might this effect the observed results?

4. Ni-NTA, additionally introduced a positive charge to the membrane. Are all Ni-NTA molecules saturated, or does the membrane carry a net positive charge. If so, could this lead to additional charge interactions between the membrane and the protein?

5. How did the authors determine the concentration of Cdh23 to use within the GUV and how does this concentration compare to that commonly observed within cell? Would sufficiently high concentration of Cdh23 on the GUV membrane lead to domain formation i.e. a 2D analogue to the droplets observed in solution?

6. What is meant by the term core and periphery of GUV? Please clarify.

7. The authors make use of a number of different cell lines to assess the influence of Cdh23 expression on cell adhesion. It is unclear if the expression of other adhesion molecules which could contribute towards cell adhesion are equally expressed by the chosen cell lines. Consequently it is hard to confidently state that the observed differences are due to Cdh23 expression alone. Could the authors confirm this results by differentially transducing a single cell line with Cdh23? Via a cell sorting approach, populations of low, and high Cdh23 cells could be generated and used to assess the effects of its expression on cell adhesion.

Reviewer #2 (Remarks to the Author):

The author claims that cadherin-23, a long non-classical cadherin, can form lateral-clustering, albeit via transient, weak, but multivalent and spatially distributed hydrophobic interactions. The

author further claims that while no-specific cis-dimer interaction is required for the clustering, cis-clustering (at least at in cellulo level), can accelerates cell-to-cell adhesion and strengthening the cells junctions. Lastly, the author claims that M2-macrophages expresses cadherin-23 undergo fast attachment to circulatory tumor cells during metastasis.

The usage of 1,6HD as to weaken hydrophobic interactions is clearly showing the hydrophobic requirement for the interactions of cadherin-23. The author validate first using in cellulo model before testing with cell lines. Overall, certain claims at least at cellulo and in vitro level are proper, data presentation needs to be revised, especially when explaining about the cancer cell lines. While the author acknowledge that physiological clustering with rapid adhesion is not yet explored, the author do claims that M2-macrophages have cadherin-23 and helps in attachment to CTCs. However, this claim is not founded by proper data either in vitro or in vivo. Primarily, in the discussion, the author claim 'though-speculative, the fast adhesion between M2 macrophages and CTCs is facilitated by the condensed Cdh23 droplets'. While the other papers referenced by the author do show Cdh23 overexpression in M2 macrophages, no evidence by the references or the author explain the association between M2 maccrophages and CTC and how the condensed Cdh23 droplets help in the fast adhesion between M2 macrophages and CTC. In summary, certain claims need to be shown by better clarity or better controls/evidences.

Figure 1B: ladder to show kDA of PAGE is not labelled to identify monomer vs dimer size. Also, the band for monomer is very faint that its questionable whether the band is actually background or not.

Video B and Fig 1C and F: Resolution is quite low that its difficult to identify, as locations are different in location to the other pictures shown.

Fig. S2: Author claim that fusion events are in range with other proteins that undergo LLPS, but did not actually show comparison with 'other proteins'. Fig S2 only show about cadherin-23.

Fig 3 Does the HEK293 actually express cadherin-23, and if so, at what level of expression. A simple flow-cytometry experiment, if there is an antibody available, would be beneficial.

Fig S4. Data that forms why HEK293 is used for Fig 3. It might be better to put this before fig 3 to explain why the author used HEK293 for fig 3.

Fig S5. This should be included in the main figure, to differentiate aggregation between cell lines that do have cadherin-23 and lines that do not.

There is no data provided by author that can prove the claim of the author that the M2-macrophages do express cadherin-23 and can attached fast to circulatory tumor cells.

Reviewer#1

1. The authors state in the text that the "The phase-separated state is stable over a range of ionic strength", however no figure is referenced that shows this specific data. Please include the reference in the revised text.

Response: We apologize for not referencing the figure for this particular data. We have incorporated the figure reference, **Figure 1H**, in the revised manuscript (page 4, the Results section).

"The phase-separated state is stable over a range of ionic strength (**Fig. 1H**)."

2. In figure H,I,J, the dependence of droplet growth on NaCl, and Ca²⁺ concentration is shown. Given the previous statement about droplet stability and ionic strength, there actually seems to be a quite complex relationship between droplet size and ionic strength. What is the origin of this relationship, especially considering the conclusion that hydrophobic and not electrostatic interactions are driving the weak interactions between Cdh23?

Response: We appreciate the careful and detailed comments by the reviewer. Beyond a critical ionic strength of 235 mM, the liquid droplets of Cdh23 grow monotonously with increasing ionic strength and reach to a maximum. Beyond maxima, the droplets shrink in size again with increasing ionic strength. Such a biphasic relation is due to the interplay between hydrophobic and electrostatic interactions. An array of LLPS experiments with droplet-disrupting agents (ATP and 1,6-HD) indicated that hydrophobic interactions predominantly induced liquid-liquid phase separation of Cdh23 (**Figures 1 D and E in the main manuscript**). Cdh23 is a giant elongated protein with a molecular weight of 350 kDa and possesses several electrostatic as well hydrophobic interactome sites across. It was thus expected that the ionic strength of the buffer would have a pivotal role in the LLPS of Cdh23. The 'no show of droplets' at the ionic strengths lower than 235 mM is likely due to dominating electrostatic repulsions over hydrophobic interactions (**Figure 1H in the main manuscript**). The higher propensities of LLPS in the range of 235 mM to 1035 mM could be due to the optimization of the hydrophobic interactions over the screening of electrostatic repulsions by ions. Following the same trend, we noticed solid condensates of Cdh23 beyond the ionic strength of 1035 mM.

The role of calcium ions is not limited to the screening of electrostatic charges; additionally, they modulate protein conformations. Cadherins are identified as calcium-binding proteins to mediate all their canonical functions. The structural studies showed that at higher calcium levels, the proteins become more rigid and obtain certain orientations and orderedness that facilitate the interactions of proteins ¹.

3. The authors use a minimal membrane system to investigate cis interactions with the 2D membrane environment, coupling the Cdh23 to the GUV membrane using Ni-NTA lipids. Can the authors comment on the orientation of the Cdh23? Presumably, given the lack of transmembrane domains, the molecule would have more degrees of freedom, increasing the likelihood of interactions with the membrane that may lead to the molecule lying flat on the membrane. Can the authors rule this out and how might this effect the observed results?

Response: We thank the reviewer for raising the interesting point. We have addressed it from the following experiments.

The protein constructs used here possess 6xHis and eGFP tags at the C-terminus, whereas GUVs are constituted by a mixture of DOPC and DGS Ni NTA lipids. 6xHis-tag of proteins at C-terminus thus attach to surface-bound Ni²⁺ ions via coordination chemistry. To estimate the non-specific attachment of proteins onto the GUV membrane, we performed a control experiment with GUVs composed of only DOPC lipids. We allowed proteins to attach to GUVs nonspecifically and monitored the extent of attachment from the fluorescence imaging of eGFP (**Figure R1**). The absence of any eGFP on GUV surfaces is indicative of undetectable non-specific attachment of proteins on GUVs. This figure is now incorporated in the revised supplementary file and referred to in the Methods section.

Further, to monitor the extent of lateral-clustering of Cdh23 on GUVs, we titrated the Cdh23 EC1-27 attached GUVs with imidazole and monitored the change in clustering from the diffusion-rate. Washing with increasing concentration of imidazole, leached Cdh23 gradually from GUVs, further indicating specific and terminal attachment of Cdh23 to GUVs (**Figure 4C in the main manuscript**).

The final set of evidence of specific C-terminal attachment of Cdh23 on GUVs is demonstrated from the GUV aggregations. Notably, Cdh23 mediates trans-homophilic interactions among neighboring cells at high-Ca²⁺. Two N-terminal EC domains engage in the trans-interactions. Ca²⁺ ions maintain a relatively stiff protein-conformations and make the terminal domains accessible for interactions. Accordingly, we noticed aggregations of Cdh23-attached GUVs at higher calcium contents (3 mM to 6 mM), indicating the orientation of proteins with N-termini freely accessible for interactions. (**Figure R2 or Figure S5 A in the revised supplementary file**). However, no aggregation was observed at lower calcium levels (<1 mM). The proper orientation of proteins exposing the N-termini is vital for such interactions between Cdh23 molecules. Notably, we did not notice the aggregation of GUVs not attached with Cdh23 at higher and lower calcium levels. Collectively, these observations indicate that protein specifically binds to GUV membranes through the C-terminus region. The protein's N-terminus region is available for the interactions, and the protein is unlikely to lie flat on the membrane.

Figure R1 (Figure S4 in the revised supplementary file). The specific attachment of Cdh23 to GUV membranes. The representative phase contrast and fluorescence (Nile Red and Cdh23-eGFP) images of DOPC (upper row) and DOPC+DGS Ni NTA GUVs (lower row). Scale bar: 15 μm .

Figure R2 (Figure S5 A in the revised supplementary file). The representative fluorescence images show the aggregation of GUVs specifically attached with Cdh23 over time. Scale bar: 20 μm .

4. Ni-NTA, additionally introduced a positive charge to the membrane. Are all Ni-NTA molecules saturated, or does the membrane carry a net positive charge. If so, could this lead to additional charge interactions between the membrane and the protein?

Response: As explained in the previous response, we have shown the aggregation of Cdh23-attached GUV membranes at higher calcium levels (3 mM to 6 mM) in the buffer (**Figure R3 A**). Chelating the calcium ions out from proteins using 1,2-bis(2-Aminophenoxy)ethane-N,N,N',N'-tetra acetate (BAPTA) disrupted the GUV aggregates, re-instating that specific protein-protein interactions responsible for GUV aggregation (**Figure R3 B**). In the control experiment, where the buffer was added to GUV aggregates showed no effect on their aggregation. A similar result was observed with the Cdh23-labeled beads subjected to calcium chelation². These results indicate that the proteins are available to mediate the interactions between GUVs and not interact nonspecifically with Ni²⁺ ions on the membrane.

Figure R3 (Figure S5 in the revised supplementary file). (A) The representative fluorescence images of GUVs specifically attached with Cdh23 undergoing aggregation over time. (B) The representative fluorescence images of GUVs specifically attached with Cdh23 detaching from each other in the presence of a calcium chelator, BAPTA. Scale bar: 20 μ m.

5. How did the authors determine the concentration of Cdh23 to use within the GUV and how does this concentration compare to that commonly observed within cell? Would sufficiently high concentration of Cdh23 on the GUV membrane lead to domain formation i.e. a 2D analogue to the droplets observed in solution?

Response: The concentration of proteins on GUVs is limited by the percentage of DGS Ni NTA lipids in GUVs. We used 2.5% of DGS Ni NTA and 97.5% DOPC lipids and formed GUVs. We next varied the protein concentrations during incubation with GUVs and monitored the protein attachment from fluorescence imaging. Our *in solution* experiments indicated that Cdh23 undergoes liquid-liquid phase separation in solution at a minimum protein concentration of 2.5 μM (**Figure 1 H in the main manuscript**). Hence, we considered the concentration of Cdh23 higher than 2.5 μM while incubating it with GUV membranes and obtained the best contrasting green fluorescent boundaries of GUVs at a protein concentration of 6.0 μM . We, thereafter, incubated GUVs with 6.0 μM of proteins throughout the experiments unless mentioned.

To monitor the extent of clustering with the concentration of proteins on GUVs, we gradually leached out Cdh23 proteins from GUV membranes using varying concentrations of imidazole. The variations in the clusters were monitored using FRAP (**Figure 4 C in the main manuscript**). We observed that the change in protein density does alter the clusters.

The density of cadherins on cell membranes varies as per the regulations in protein-expression. We identified five cancer cells that express Cdh23 differentially and, thus, expected to mediate cis-clustering of different extents (**Figure S6 A in the revised supplementary file**). Differences in cis-clustering affect the cell-adhesion kinetics as described in the manuscript (**Figures 3 C and D in the revised manuscript**).

6. What is meant by the term core and periphery of GUV? Please clarify.

Response: Our apologies for referring 'pole' as 'core' in the manuscript. We performed the FRAP experiments at the polar and periphery regions of GUVs. We have mistakenly written it as the core instead of the pole. Though, we had correctly mentioned it as the polar region in the materials and methods section. We have included the images showing the pole and periphery regions of GUV and corrected the word "core" to "polar" in the revised manuscript as shown below (**Figure R4**). Since we observe no striking differences in the diffusion coefficients between pole and periphery, we would like to remove the periphery data from the manuscript if reviewers permit. Removal of the periphery data will avoid any confusion related to the regions.

"We performed the FRAP experiments at the **polar** region of GUVs for all variants of Cdh23 and recorded the slowest diffusion-rate of $0.124 \pm 0.005 \mu\text{m}^2 / \text{s}$ for Cdh23 EC1-21 and the fastest for Cdh23 EC1-10 is $0.60 \pm 0.025 \mu\text{m}^2 / \text{s}$ (Table 2). Also, we performed FRAP experiments at the periphery region of GUVs and noticed no significant differences in diffusion rates between **polar** and periphery regions (Table S1)."

Figure R4. The representative fluorescence images show the polar (**A**) and periphery (**B**) regions of GUV.

7. The authors make use of a number of different cell lines to assess the influence of Cdh23 expression on cell adhesion. It is unclear if the expression of other adhesion molecules which could contribute towards cell adhesion are equally expressed by the chosen cell lines. Consequently it is hard to confidently state that the observed differences are due to Cdh23 expression alone. Could the authors confirm this results by differentially transducing a single cell line with Cdh23? Via a cell sorting approach, populations of low, and high Cdh23 cells could be generated and used to asses the effects of it's expression on cell adhesion.

Response: We thank the reviewer for raising this genuine concern and providing a valuable suggestion. We too had this concern. In order to address the concern, we estimated the mRNA expression levels of classical cadherins, E-cadherin and N-cadherin, in the mentioned cell lines using qRT-PCR (**Figure R5**). Similar to the expression patterns of Cdh23, A549 and HaCaT have comparable expressions of E-cadherin. However, A549 expresses N-cadherin orders higher than HaCaT. Expression of N-cadherin in HEK293 and HaCaT are comparable. Cell-adhesion kinetics of A549 and HaCaT are comparable too, indicating that at least N-cadherin may not be regulating the kinetics. To verify the effect of Cdh23 in cell-adhesion kinetics, we silenced the expression of Cdh23 in A549 cells using Cdh23-siRNA, and compared cell-adhesion kinetics with untransfected cells. **Figures 4 A and B** indicate that silencing Cdh23 does dampen the kinetics significantly, indicating the active role of Cdh23 in cell-adhesion kinetics. To note, treatment of Cdh23-siRNA does not alter the E-cadherin expression (**Figure R6**).

Further, we used HEK293 cells over-expressing Cdh23 and compared the cell-adhesion kinetics with non-transfected and Cdh23-silenced cell lines (**Figures 3 A and B in the revised manuscript**). Down-regulation of Cdh23 did not allow the cell-aggregation for HEK293 cells treated with Cdh23-siRNA (**Figure 4 A and B in the revised manuscript**).

Figure R5. mRNA level expression of E-cadherin (**A**) and N-cadherin (**B**) was measured in different cell lines using qRT-PCR.

Figure R6 (Figure S7 in the revised supplementary file). Cdh23 and E-cadherin mRNA expressions were measured in A549 cells silenced with Cdh23 and control cells using qRT-PCR.

As suggested by the reviewer, we did sort the high-Cdh23 and low-Cdh23 expressing HEK293 cells using flow cytometry. With a very low transfection efficiency of such a giant plasmid, sorted cells were very less in numbers, thus stressed, and the majority of the cells lead to death while performing aggregation assays, recurrently. Though we agree with the reviewer that the

aggregation assay with sorted cells would have been an appropriate experiment to claim the differences in aggregation due to Cdh23 expression, unfortunately we failed to get any satisfactory results. Moreover, we believe that we have now provided enough evidence to justify the role of Cdh23 and its cis-clusters exclusively on the kinetics of cell-cell adhesion.

Reviewer #2

1. Figure 1B: ladder to show kDA of PAGE is not labeled to identify monomer vs dimer size. Also, the band for monomer is very faint that its questionable whether the band is actually background or not

Response: We apologize for not labeling the bands in the ladder. We have incorporated a new SDS PAGE image in **Figure 1B** in the revised version and marked the bands in the ladder (**Figure R7**). The previous PAGE was not clear enough to make the assessment. Even in the new PAGE, a bubble can be noticed at the center of the PAGE. We could not remove the bubble as the gel is formed by 5% acryl/bisacrylamide, it is soft and fragile, thus making it difficult to handle. We need to use only 5% acryl/bisacrylamide as the protein is giant with a MW of >315 kDa.

Figure R7 (Figure 1 in the revised manuscript).

2. Video B and Fig 1C and F: Resolution is quite low that its difficult to identify, as locations are different in location to the other pictures shown.

Response: We apologize to the reviewer for including the low-resolution images and video. We have replaced the images in **Figures 1C and F (Figure R7)** and enhanced the contrast of the **Video A** in the revised manuscript.

3. Fig. S2: Author claims that fusion events are in range with other proteins that undergo LLPS, but did not actually show comparison with 'other proteins'. Fig S2 only shows cadherin-23.

Response: The time of fusion events in numeric values is rarely reported in the literature. However, when we analyzed the fusion videos published for the respective proteins, the approximate time for fusion was found in the range of 2-4 s. The representative data for proteins including α -Synuclein, FUS, Prion, L-SEPA, and PGL showed the fusion events occurring in 2 - 4 s time scale³⁻⁶. The Fusion event for Cdh23 droplets (Fig. S2) took place in a 4 s time scale too. We have compared the time scales as a part of the internal analysis.

4. Fig 3 Does the HEK293 actually express cadherin-23, and if so, at what level of expression. A simple flow-cytometry experiment, if there is an antibody available, would be beneficial.

Response: We have performed qRT-PCR to quantify the mRNA level expression of Cdh23 in different cell lines, including HEK293, HeLa, HaCaT, and A549. The human embryonic kidney, HEK293 cells, showed a low level of expression of Cdh23 compared to lung cancer cell line A549. The data is included in supplementary **Figure S6 A**.

5. Fig S4. Data that forms why HEK293 is used for Fig 3. It might be better to put this before fig 3 to explain why the author used HEK293 for fig 3.

Response: We sincerely thank the reviewer for the suggestion. HEK293 cells express low levels of epithelial and mesenchymal markers⁷ and are one of the widely used model cell lines to study the functions of membrane proteins in cell biology⁸. Further, these cells are relatively easy to transfect over other cell lines, and better transfection efficiency was achieved even with a giant plasmid bearing Cdh23⁹. We have used HEK293 cells to verify the effect of lateral clustering of Cdh23 on cell-cell adhesion because the endogenous expression of Cdh23 in HEK293 is not significant. Accordingly, we transfected Cdh23 variants, Cdh23 EC1-10 and Cdh23 EC1-27, in HEK293 cells and extended the cell-aggregation experiments.

We have cited the supplementary **Figure S4** before main **Figure 3** and included the following text in the revised manuscript.

"HEK293 is one of the widely used cell lines to study the functions of membrane proteins in cell biology. Further, these cells are relatively easy to transfect over other cell lines, and high transfection efficiency can be achieved. We have used HEK293 cells to verify the effect of lateral

clustering of Cdh23 on cell-cell adhesion because the endogenous expression of Cdh23 in these cells is not significant. "

6. Fig S5. This should be included in the main figure, to differentiate aggregation between cell lines that do have cadherin-23 and lines that do not.

Response: Our sincere thanks to the reviewer for the suggestion that improves the data presentation. We combined supplementary **Figure S5** with main **Figure 3** and included it (**Figure R8**) in the revised manuscript.

Figure R8 (Figure 3 in the revised manuscript). Lateral clustering of Cdh23 on the membrane enhances the kinetics of cell-cell adhesion. (A) Time-stamp bright-field images of cell-cell aggregations of HEK293 cells untransfected (1st column), transfected with Cdh23 EC1-10 (2nd column), Cdh23 EC1-27 (3rd column), and Cdh23 EC1-27 along with the treatment with 1,6-HD (4th column). Scale bar: 50 μ m. (B) The time-dependent growth of the cell-cell aggregation area of HEK293 cells exogenously expressing Cdh23 EC1-27 (red), Cdh23 EC1-27 and treated with 1,6-HD (olive), Cdh23 EC1-10 (blue), and untransfected cells (black). The error bars represent the standard error of the mean (SEM) with N=15 aggregates. (C) Time-stamp bright-field images of cell-aggregates of HeLa, HEK293, HaCaT, and A549 cells differentially expressing endogenous Cdh23. Scale bar: 50 μ m. (D) Growth of cell-cell aggregation area (in

μm^2) with time for HeLa (red), HEK293 (Olive), HaCaT (black), and A549 (blue) cell lines. Error bars represent the standard error of the mean (SEM) for N=15 aggregates.

7. There is no data provided by author that can prove the claim of the author that the M2-macrophages do express cadherin-23 and can attach fast to circulatory tumor cells.

Response: We appreciate the opportunity to include additional data. We have performed a qRT-PCR experiment with RAW264.7 macrophage cells (M0) polarized to M1 (anti-tumorigenic) and M2 (tumor-associated) types. The data showed the up-regulated expression of Cdh23 in M2-type macrophages compared to un-polarized (M0) cells (**Figure R9**). We included this figure in the revised supplementary file.

The crosstalk between tumor-associated M2 macrophages and circulating tumor clusters (CTCs) is reported^{10,11}. The role of cell-surface proteins like intercellular adhesion molecule 1 (ICAM1) to mediate the adhesion between macrophages and tumor cells and in the formation of CTCs is well understood¹². In the same context, we speculate that overexpressed Cdh23 on M2 macrophages may facilitate the faster adhesion between M2-macrophages and CTCs. However, this needs to be experimentally verified.

The fast attachment of M2 with CTCs is though speculative at this stage, however, physiologically relevant. We changed the statement made in the abstract accordingly to avoid any mis-interpretation.

Figure R9 (Figure S8 in the revised supplementary file). qRT-PCR estimated that Cdh23 expression is down-regulated in anti-tumorigenic (M1) macrophages (**A**) and up-regulated in tumor-associated (M2) macrophages (**B**). The un-polarized RAW264.7 cells (M0) were polarized to M1 using Lipopolysaccharide and Interferon gamma cytokines, and Interleukin-4 (IL-4) cytokines were used to polarize RAW264.7 cells (M0) to M2.

References:

- 1 Cailliez, F. & Lavery, R. Cadherin mechanics and complexation: the importance of calcium binding. *Biophysical journal* 89, 3895-3903, doi:10.1529/biophysj.105.067322 (2005).
- 2 Singaraju, G. S. *et al.* Structural basis of the strong cell-cell junction formed by cadherin-23. *The FEBS journal*, doi:10.1111/febs.15141 (2019).
- 3 Agarwal, A., Arora, L., Rai, S. K., Avni, A. & Mukhopadhyay, S. Spatiotemporal modulations in heterotypic condensates of prion and alpha-synuclein control phase transitions and amyloid conversion. *Nat Commun* 13, 1154, doi:10.1038/s41467-022-28797-5 (2022).
- 4 Patel, A. *et al.* A Liquid-to-Solid Phase Transition of the ALS Protein FUS Accelerated by Disease Mutation. *Cell* 162, 1066-1077, doi:10.1016/j.cell.2015.07.047 (2015).
- 5 Agarwal, A., Rai, S. K., Avni, A. & Mukhopadhyay, S. An intrinsically disordered pathological prion variant Y145Stop converts into self-seeding amyloids via liquid-liquid phase separation. *Proceedings of the National Academy of Sciences of the United States of America* 118, doi:10.1073/pnas.2100968118 (2021).
- 6 Wang, Z., Zhang, G. & Zhang, H. Protocol for analyzing protein liquid-liquid phase separation. *Biophysics Reports* 5, 1-9, doi:10.1007/s41048-018-0078-7 (2019).
- 7 Inada, M., Izawa, G., Kobayashi, W. & Ozawa, M. 293 cells express both epithelial as well as mesenchymal cell adhesion molecules. *International journal of molecular medicine* 37, 1521-1527, doi:10.3892/ijmm.2016.2568 (2016).
- 8 Lin, C. Y. *et al.* Enhancing Protein Expression in HEK-293 Cells by Lowering Culture Temperature. *PloS one* 10, e0123562, doi:10.1371/journal.pone.0123562 (2015).
- 9 Ooi, A., Wong, A., Esau, L., Lemtiri-Chlieh, F. & Gehring, C. A Guide to Transient Expression of Membrane Proteins in HEK-293 Cells for Functional Characterization. *Frontiers in physiology* 7, 300, doi:10.3389/fphys.2016.00300 (2016).
- 10 Amintas, S. *et al.* Circulating Tumor Cell Clusters: United We Stand Divided We Fall. *International journal of molecular sciences* 21, doi:10.3390/ijms21072653 (2020).
- 11 Osmulski, P. A. *et al.* Contacts with Macrophages Promote an Aggressive Nanomechanical Phenotype of Circulating Tumor Cells in Prostate Cancer. *Cancer research* 81, 4110-4123, doi:10.1158/0008-5472.CAN-20-3595 (2021).
- 12 Taftaf, R. *et al.* ICAM1 initiates CTC cluster formation and trans-endothelial migration in lung metastasis of breast cancer. *Nat Commun* 12, 4867, doi:10.1038/s41467-021-25189-z (2021).

REVIEWERS' COMMENTS:

Reviewer #1 (Remarks to the Author):

The authors have largely addressed my initial concerns, performing new experiments and including a number of new figures within the manuscript.

I would like to thank the authors for provided new figures supporting their initial conclusions regarding the orientation of the Cdh23 on the membrane.

Regarding the relative expression of adhesion molecules on the different cell lines used, the authors have included new data using RNA-seq and indeed attempted to sort cells to normalise differences in expression. While these efforts certainly add to the manuscript, I would suggest that the authors included an extended discussion of the potential influence of differing expression of other adhesion molecules between cell lines as a possible confounding factor within their experiments.

Reviewer #2 (Remarks to the Author):

Thank you for the revision of the manuscript. I am pleased with the revision made by the authors thus far.

Below are the detailed comments for each rebuttal to my previous comments:

1. Response to rebuttal 1: Thank you for incorporating new image with indicated bands in the ladder. Thank you for clarifying the restriction of using only 5% acrylic/bisacrylamide due to the MW of the protein. The new image is indeed better than the previous.
2. Response to rebuttal 2: Thank you for the replacement of the images and making better enhancement of the video contrast.
3. Response to rebuttal 3: Thank you for the clarification. If the referencing have not been included in the manuscript text with regards to Fig S2, please do so. I believe this will make better clarification to readers.
4. Response to rebuttal 4: Thank you for the clarification. I believe the association between Fig 3 and Fig S6A has been made in the manuscript text, if not, it would be beneficial to add into the manuscript text, for ease of reading.
5. Response to rebuttal 5: Thank you for making the adjustment in the citing within the manuscript text. It now reads well.
6. Response to rebuttal 6: Thank you for making the adjustment. It reads well now.
7. Response to rebuttal 7: Thank you for the additional data to clarify the M1 and M2 vs M0 of Cdh23. Thank you for including the reference. If the referencing have not been included in the manuscript text, please do so, as I believe this will make better clarification to readers.

Reviewer#1

1. Regarding the relative expression of adhesion molecules on the different cell lines used, the authors have included new data using RNA-seq and indeed attempted to sort cells to normalise differences in expression. While these efforts certainly add to the manuscript, I would suggest that the authors included an extended discussion of the potential influence of differing expression of other adhesion molecules between cell lines as a possible confounding factor within their experiments.

Response: We thank the reviewer for providing the valuable suggestion. Accordingly, we have added the following text to the results section in the manuscript (Page 8).

“Additionally, we estimated the mRNA expression levels of two classical cadherins, E-cadherin and N-cadherin, in the mentioned cell lines using qRT-PCR (**Supplementary Fig. 9**). Similar to the expression patterns of Cdh23, A549 and HaCaT have comparable expressions of E-cadherin. However, A549 expresses N-cadherin orders higher than HaCaT. Expression of N-cadherin in HEK293 and HaCaT are comparable. Cell-adhesion kinetics of A549 and HaCaT are comparable too, indicating that at least N-cadherin may not be regulating the kinetics. To verify the effect of Cdh23 in cell-adhesion kinetics, we silenced the expression of Cdh23 in A549 cells using Cdh23-siRNA. The silencing of Cdh23 does dampen the kinetics significantly, indicating its active participation (**Fig. 4a, b**).”

Reviewer #2

1. Response to rebuttal 1: Thank you for incorporating new image with indicated bands in the ladder. Thank you for clarifying the restriction of using only 5% acrylic/bisacrylamide due to the MW of the protein. The new image is indeed better than the previous.

2. Response to rebuttal 2: Thank you for the replacement of the images and making better enhancement of the video contrast.

Response: We have verified the images and video in comments 1 and 2 and reconfirmed.

3. Response to rebuttal 3: Thank you for the clarification. If the referencing have not been included in the manuscript text with regards to Fig S2, please do so. I believe this will make better clarification to readers.

Response: We have included references 28, 29, and 30 in the main manuscript related to Fig S2 (Supplementary Fig. 2 in the current version).

4. Response to rebuttal 4: Thank you for the clarification. I believe the association between Fig 3 and Fig S6A has been made in the manuscript text, if not, it would be beneficial to add into the manuscript text, for ease of reading.

5. Response to rebuttal 5 & 6: It is done.

Response: We have verified our responses for comments 4, 5, and 6 and reconfirmed them.

7. Response to rebuttal 7: Thank you for the additional data to clarify the M1 and M2 vs M0 of Cdh23. Thank you for including the reference. If the referencing have not been included in the manuscript text, please do so, as I believe this will make better clarification to readers.

Response: We have included references 47 and 48 in the main manuscript related to additional data to clarify M1 and M2 vs. M0.

We have removed 'Conclusion'.

In the current manuscript, we modified the acknowledgments section.